# Bioengineering Composite Aerogel-Based Scaffolds That Influence Porous Microstructure, Mechanical Properties and In Vivo Regeneration for Bone Tissue Application

**DOI:** 10.3390/ma16124483

**Published:** 2023-06-20

**Authors:** Mariana Souto-Lopes, Maria Helena Fernandes, Fernando Jorge Monteiro, Christiane Laranjo Salgado

**Affiliations:** 1i3S—Instituto de Investigação e Inovação em Saúde da Universidade do Porto, 4200-135 Porto, Portugal; marianasoutolopes@gmail.com (M.S.-L.); fjmont@fe.up.pt (F.J.M.); 2INEB—Instituto de Engenharia Biomédica, Universidade do Porto, 4200-135 Porto, Portugal; 3Departamento de Engenharia Metalúrgica e de Materiais, Faculdade de Engenharia da Universidade do Porto, 4200-465 Porto, Portugal; 4Bonelab–Laboratory for Bone Metabolism and Regeneration, Faculdade de Medicina Dentária da Universidade do Porto, 4200-393 Porto, Portugal; 5LAQV/REQUIMTE—Laboratório Associado para a Química Verde/Rede de Química e Tecnologia, 4169-007 Porto, Portugal; 6Porto Comprehensive Cancer Center (P.CCC), R. Dr. António Bernardino de Almeida, 4200–072 Porto, Portugal

**Keywords:** aerogels, bone regeneration, composite scaffolds, porous microstructure, mechanical properties, biomaterials

## Abstract

Tissue regeneration of large bone defects is still a clinical challenge. Bone tissue engineering employs biomimetic strategies to produce graft composite scaffolds that resemble the bone extracellular matrix to guide and promote osteogenic differentiation of the host precursor cells. Aerogel-based bone scaffold preparation methods have been increasingly improved to overcome the difficulties in balancing the need for an open highly porous and hierarchically organized microstructure with compression resistance to withstand bone physiological loads, especially in wet conditions. Moreover, these improved aerogel scaffolds have been implanted in vivo in critical bone defects, in order to test their bone regeneration potential. This review addresses recently published studies on aerogel composite (organic/inorganic)-based scaffolds, having in mind the various cutting-edge technologies and raw biomaterials used, as well as the improvements that are still a challenge in terms of their relevant properties. Finally, the lack of 3D in vitro models of bone tissue for regeneration studies is emphasized, as well as the need for further developments to overcome and minimize the requirement for studies using in vivo animal models.

## 1. Introduction

### 1.1. Biomimetic Graft Biomaterials for Bone Regeneration

Bone is a highly specialized hard connective tissue with a macroscopic and microscopic hierarchical structure that is continuously being remodeled through the concerted activities of bone-forming osteoblasts and bone-resorbing osteoclasts [1]. As schematized in Figure 1, macroscopically, bone encompasses a dense outer layer, the cortical or compact bone, which surrounds the inner highly porous (total porosity of 40 to 90% [2]) marrow cavity formed by trabecular or cancellous bone [3,4]. The microscopic arrangement of mature cortical bone is composed of the Harversian canal systems (osteons) surrounded by parallel or concentric lamellar systems [3,4]. On the other hand, trabeculae and spicules of cancellous bone are extensions of the inner circumferential lamellae of cortical bone into the marrow space, but their lamellar configuration is irregular and does not have osteons [3,4]. While the lamellae of compact bone enclose a high number of dispersed lacunae, each with an osteocyte and a network of interconnecting canaliculi to maintain cell signaling and bone metabolism, in cancellous bone, the osteocytes inside the lamellae perform metabolic exchanges by diffusion from the bone marrow [3,4].

Osteocytes, the more abundant bone cells, are the mature cells that result from osteoblasts becoming trapped in their own lacunae upon calcification of the extracellular matrix (ECM) [3,6,7]. Bone ECM is composed of an organic phase, 30% *w*/*w*, mostly of collagen type I crosslinked fibers secreted by the osteoblasts. On top of these fibers, hydroxyapatite nanocrystals are orderly deposited which, together with amorphous calcium phosphate, comprise the inorganic phase of bone (70% *w*/*w*) [2,8,9]. The combination of both organic and inorganic phases decisively contributes to the simultaneous hardness and flexibility of bone, making it able to withstand the forces exerted by physiologic body functions [2,10,11].

Though bone is a dynamic tissue that is able to continuously remodel, and also to repair itself in case of injury, some traumatic and pathologic lesions are so extensive that they exceed the organism’s regeneration capacity [12,13,14,15,16]. Moreover, with an increasingly aging world population, and increased demand for higher health standards and quality of life, it has become imperative to develop strategies to overcome bone regeneration obstacles [12,17,18,19]. Since there are limitations to the use of autografts and allografts for surgical filling of large bone void defects [14,15,16,18,19,20], in recent decades, bone tissue engineering has been developing biomimetic alloplastic graft biomaterials inspired by the above-mentioned macro and microstructure of bone, as well as by its ECM physicochemical features [2,11,13,14,16,21,22].

From a materials science point of view, bone ECM can be considered a composite due to its organic/inorganic synergistic nature [8]. Therefore, efforts have been made to develop bone substitute materials that combine an organic flexible polymeric matrix reinforced by an inorganic dispersed phase [7,8,9,10,20]. Furthermore, the raw materials chosen for the bone graft composition should be biocompatible, as well as bioactive to stimulate osteoprogenitor cells to migrate and differentiate into osteoblasts that will secrete collagen for the new bone tissue [16,18]. Moreover, the structure and texture of these graft scaffolds should display high hierarchical open porosity, with meso (nano), micro and macropores (100–500 μm) to allow the migration and attachment of osteoblasts, as well as the ingrowth of new blood vessels [7,15,18,20,23] and the circulation of oxygen, nutrients, metabolites and signaling molecules, and consequently a large inner hydrophilic surface area to allow water uptake, protein adsorption and ion exchange [23,24]. Moreover, bone graft scaffolds should exhibit mechanical properties comparable to native bone (cancellous bone demonstrates compressive strengths ranging between 7 and 10 MPa [25], ultimate tensile strength near 1.2 MPa and bending strength around 8 MPa [9]), which can be promoted by the existence of nanosized pores, to reach ideally 5–10 MPa of compressive strength [26].

This review offers an up–to–date overview of the advances in aerogel-based composite biomaterials that provide biomimetic structures for bone regeneration. Recently published data (last 5 years) of aerogel composite scaffolds for bone regeneration are analyzed, and the outcomes are compared in terms of the scaffolds’ different production methods, porous microstructure, mechanical properties and in vivo bone regeneration capacity.

### 1.2. Aerogel-Based Biomaterials

Though the number of scientific publications on aerogel-based materials has been increasing exponentially in the last years (Figure 2a), it is in fact a concept that goes back to 1931, when Kistler produced a highly porous solid material after replacing the liquid phase from a gel by air through supercritical conditions with minimal volume shrinking [27,28] (usually below 25%) [28]. Since then, aerogel materials have been developed and applied to a wide variety of scientific and technological areas, e.g., civil engineering, electrical engineering, aerospace engineering, chemical engineering and environmental applications, due to their remarkable physicochemical properties [29,30]. Most of these materials are silica-based [29] and zeolites [31].

According to IUPAC, aerogel is a “gel comprised of a microporous solid in which the dispersed phase is a gas” [31], and additionally exhibits features such as high porosity (>80%), open pore network, high mesoporosity [32], low bulk density (<0.5 g·cm^–3^), high specific surface area (500–1200 m^2^·g^−1^), low thermal conductivity, low speed of sound propagation and low dielectric constant [26,29,33,34,35]. More recently, due to these outstanding characteristics, as well as the possibility of using a variety of raw materials in their composition [32,36], aerogels are also being applied in the biomedical and pharmaceutical fields, namely as scaffolds for tissue regeneration, wound healing dressings, carriers for drug delivery, biosensors for diagnostics [29,33,34,36,37], antimicrobial activity agents and decontaminating compounds [34]. Accordingly, as observed in Figure 2b, aerogel-based scaffolds are being developed specifically for bone tissue regeneration, resorting to organic natural (silk fibroin, chitosan, cellulose, starch, alginate) [14,15,17,24,32,38] or synthetic polymers (poly(e–caprolactone) (PCL)) [17] and inorganic compounds (cellulose nanocrystals, β–Tricalcium phosphate, silica, hydroxyapatite) [18,20,36,39], or a combination of both (composite/hybrid) (e.g., silica–silk fibroin) [11,30,33].

## 2. Preparation Methods of Composite Aerogels for Bone Regeneration

Generally, the first step to obtain a composite aerogel is to prepare a homogeneous colloidal solution with the precursor compounds [33,37]. These precursors can be solely the monomers or precursors dissolved in water or organic solvents [35] or may already include the inorganic phase or other target components (in situ) before the sol–gel process [16,30]. At this stage, polymer gelation occurs through chemical, physical [35] or enzymatic crosslinking [40], forming a wet gel/aquagel [33] (hydrogel [32] in the case the swelling agent is water, alcogel in the case of alcohols [31]). Moreover, gelation can be ionically promoted by biopolymers such as chitosan [7,15,24] and alginate in the colloidal solution [41]. Chemical crosslinking tends to form stronger bonds than physical crosslinking [24,33,35,42].

Furthermore, often micro and nanofabrication techniques such as electrospinning and 3D printing have been employed in order to obtain aerogel scaffolds that mimic the ECM 3D morphology with tunable features [10]. Therefore, electrospinning is a versatile technique that may be applied either to the precursor (organic or inorganic) solutions [10,15,43,44,45,46] or after the sol–gel process [43]. However, the direct deposition of the nanofiber layers obtained by conventional electrospinning results in 2D densely packed nanofiber mats [16,43,44,47] that are inadequate for tissue regeneration [48] and require additional processing methods to obtain 3D scaffolds [16,43,47]. Moreover, a frequently used strategy is to cut the as-spun continuous nanofibers into shorter nanofibers [48] to incorporate into scaffolds in order to improve simultaneously compression strength and flexibility [15,43,47], as well as to enhance ECM biomimetics [15,16,43,44].

Three-dimensional printing has been rapidly developing due to the need to customize the scaffolds to the irregular shapes of the bone defects through computer-aided design (CAD) technology [21,23,39,41], in a reproducible, automated and scalable way [41]. Additionally, 3D printing enables the manufacturing of scaffolds of various compositions with complex architectures and controlled porous interconnected microstructures [11,21,23,41]. Micro-extrusion-based 3D printing is an additive technique that has been commonly used for producing hydrogels for bone scaffolds, in which adjusting the formulation of the bioink gel (viscosity and rheology) is crucial to obtain an optimized extrudability [11,41] but at the same time solid enough to maintain its tailored 3D geometry upon layer-by-layer deposition and to achieve maximum shape fidelity and printing accuracy [41].

Finally, in order to obtain a solid material, it is necessary to remove the liquid phase from the wet gel by a method that respects its porous structure, maintains its volume and prevents it from collapsing [33,34,35]. Therefore, the drying technique is considered the most critical step in aerogel preparation [30]. Furthermore, if the inorganic phase (or other target substance) is not included in the precursor solutions before the sol–gel process, it may be added to the aerogel in an additional final processing step (ex situ) by adsorption or precipitation [8], loaded by a liquid or gas phase [30].

Depending on the chosen drying technique, three types of solid porous materials have been distinguished: (1) xerogels are obtained after drying for several days at room temperature and pressure; (2) cryogels are obtained upon freezing of the wet gel, followed by sublimation of the frozen solvent crystals by freeze drying; (3) aerogels are achieved when drying is accomplished in supercritical (CO_2_, ethanol, acetone or methanol) drying conditions [30,34,35]. However, in recent years, with advances in materials science, the aerogel concept has been extended to porous materials that may be produced by a liquid phase removal (e.g., freeze drying) [2,28] and display aerogel-like properties; therefore, this strict classification of solid porous materials based solely on the drying technique may not always be applied [28]. Nevertheless, from the same gel composition it is possible to obtain materials with different final properties according to the performed drying technique, since their properties depend on the final configuration of the porous matrix [28]. Due to the particularities of supercritical drying techniques, these are the ones with the least impact on the gels’ structure for obtaining aerogels when compared with other techniques [28]. The most relevant drying techniques employed for aerogel-based scaffolds for bone regeneration are elucidated below.

### 2.1. Supercritical Drying

Supercritical drying is based on the principle that when a certain molecule is simultaneously a liquid and a gas in conditions beyond its specific critical temperature point (T_C_) and pressure (P_C_), it is in a supercritical state [49,50]. As a result, the molecule preserves the properties of a liquid (dissolution capacity) and of a gas (high diffusion and low viscosity) [49,50].

Depending on the nature of the wet gel compounds and respective solvents, two types of supercritical drying may be applied. High-temperature supercritical drying (HTSD) operates at temperatures and pressures higher than the solvent’s (usually ethanol) critical point, and this supercritical fluid is slowly extracted from the gel [28,35]. This technique has been often used to produce silica-based aerogels with large porosity and specific surface and very low densities, as well as other inorganic and metal oxide-based aerogels [35]. However, these conditions imply more operating risks and very often are not appropriate for sensitive biopolymers (e.g., polysaccharide-based polymers such as chitosan and alginate or proteins such as collagen, gelatin and silk-fibroin) [35]. Therefore, low-temperature supercritical drying (LTSD) has been introduced [35], by either dissolving the wet gel organic solvent in soluble CO_2_, raising the pressure until CO_2_ reaches the supercritical state and flushing it out [28,33] or by continuously pumping a supercritical CO_2_ (scCO_2_) flow through the wet gel [28].

CO_2_ is a fairly widely available and affordable non-inflammable gas [50,51], which is environmentally friendly and recyclable [50]. The supercritical conditions of CO_2_ are T_C_ 31.1 °C (304.25 K or 87.98 °F/near-room temperature) and P_C_ 7.39 MPa (72.9 atm or 1.071 psi or 73.9 bar), which are relatively mild conditions for sensitive biopolymers [49,50,51,52]. Moreover, scCO_2_ is inert with respect to most polymers, and since it has high density, low viscosity and no surface tension (no capillary forces), it is appropriate for producing highly porous scaffolds [33,49,51]. However, in the case of some polysaccharide-based aerogels (e.g., alginate), the solvent used is water (hydrogel) [41], which is poorly soluble in scCO_2_. Consequently, in those particular cases, an extra processing step of solvent exchange (for an organic solvent) must be performed before supercritical drying [32,33,35,38,41]. Nevertheless, LTSD usually takes longer to remove all the solvent from the aerogel than HTSD [35], which may be a drawback in terms of technological scale-up.

Table 1 summarizes the most relevant outcomes in terms of porous configuration, mechanical properties and bone regeneration in vivo for supercritical dried composite aerogels.

**Table 1 materials-16-04483-t001:** Aerogel composite scaffolds prepared by scCO_2_ drying.

References	Material	Preparation Methodology	Porous Properties	In Vitro Mechanical Properties	Bone Regeneration In Vivo (microCT Analysis)
Perez-Moreno et al. (2020) [7]	Silica (SiO_2_)/chitosan (CS 0, 4, 8, 16, 20 wt.%) (SCS) composite aerogels	Ultrasonic preparation of precursors, ultrasonic sol–gel, aging and solvent exchange, scCO_2_ drying (40 °C, 10 MPa)	Pore size (nm): 11.211 (SCS8)–14.108 (SCS16) (BJH method)	Young’s modulus (MPa): 0.66 (SCS16)–11.57 (SiO_2_)	_
Perez-Moreno et al. (2023) [53]	Chitosan (CS 8 wt.%)–silica (SiO_2_) hybrid aerogel (SCS8A)	Ultrasonic preparation of precursors, ultrasonic sol–gel, aging and solvent exchange, scCO_2_ drying (40 °C, 10 MPa)	Pore size (nm): 16.9 (SiO_2_A), 17.3 (SCS8A) (BJH method)	_	_
Reyes-Peces et al. (2023) [54]	Hybrid silica–3-glycidoxypropyl trimethoxysilane (GPTMS)–gelatin (SG) (15, 25, 30 wt.% of gelatin content)-based aerogel	One-step sol–gel (with crosslinking), scCO_2_ drying (90 bar, 40 °C)	Mean pore diameter (nm): 8.6 (SG30)–10.8 (SG15) (BJH method)	Young’s modulus (MPa): 30.81 (SG15)–78.55 (SG30) (dry); 1.65 (SG30)–3.71 (SG15) (wet). Compressive strength (MPa): 3.69 (SG15)–9.90 (SG30) (dry); 0.10 (SG30)–0.33 (SG15) (wet). Maximum compressive strain (%): 14.06 (SG30)–27.67 (SG25) (dry); 4.08 (SG30)–4.55 (SG15) (wet).	_
Iglesias-Mejuto et al. (2021) [41]	3D-printed alginate (Alg 6%)–hydroxyapatite (HA 0, 8, 16, 24 wt.%) aerogel scaffolds	Sol–gel to obtain bioinks, 3D printing of hydrogels, gelation, conversion into alcogels, scCO_2_ drying (40 °C, CO_2_ flow rate 5–7 g/min, 120 bar, 4 h)	Mean pore diameter (nm): 19 (Alg 6%, HA 0%, CaCl2 1 M)–31 (Alg 6%, HA 24%, CaCl2 1 M) (BJH method). Macropores (SEM imaging). Total porosity (%): 80.33 (Alg 6%, HA 24%, CaCl2 1 M)–88.56 (Alg 6%, HA 0%, CaCl2 1 M) (helium pycnometer)	_	_
Maleki et al. (2019) [26]	Silica–silk fibroin (SF) aerogel hybrids	One step aqueous-based sol–gel, unidirectional freeze casting (slow (33 cm/h) or rapid (66 cm/h) cooling rate until −10 °C or −196 °C), scCO_2_ drying	Porosity (%): 91 (silica–SF-10-33)–94 (silica–SF-196-33 and silica–SF-196-66) (helium pycnometer). Pore diameter (nm): 16 (silica–SF-10-66 and silica–SF-196-66)–18 (silica–SF-10-33) (BJH method). Macropore diameter (µm): 0.52 (silica–SF-10-66)–17.84 (silica–SF-196-33) (SEM imaging)	Maximum compression strength (MPa): 0.36 (silica–SF-10-66)–1.6 (silica–SF-196-33). Young’ s modulus (MPa): 4.03 (silica–SF-10-33)–7.3 (silica–SF-196-33).	Femur defect in rats—25 days New bone density of scaffold (silica–SF-196-33) implanted defect was similar to native bone

BJH—Barrett–Joyner–Halenda method; SEM—scanning electron microscopy.

### 2.2. Freeze Drying

The most commonly employed aerogel drying technique for biomedical applications (including bone tissue regeneration, as shown in Table 2) is freeze drying, due to its simplicity, low cost, being environmentally friendly, low working pressures and the fact that it does not require porogen particles, gases or flammable liquids [33,42]. Through this drying process, it is possible to obtain biomaterials with high porosity and high surface area, preserved micro and nanopore structure, with low volume shrinkage and low density [33,42].

In terms of the resulting aerogel configuration, the most critical step is the freezing stage. Most solvents freeze between −5 and −20 °C, which are quite accessible freezing temperatures [35]. However, some studies have been published describing −80 °C freezing temperatures [10,55] and −196 °C (liquid N_2_) [11,21,44], as well as tailored freezing protocols to obtain specific results, such as in the case of freeze casting or cryostructuring (directional freezing) [15,23,42,44], in order to obtain, for example, anisotropic materials [26,42]. The freezing temperature and freezing rate dictate, to a great extent, the size and distribution of the ice crystals and, consequently, the size and interconnection of the pores of the final aerogel structure [42]. The pores’ mean diameter and their interconnectivity diminish with decreasing temperature and increasing freezing rate [35,42,44].

**Table 2 materials-16-04483-t002:** Aerogel composite scaffolds prepared by freeze drying.

References	Material	Preparation Methodology	Porous Properties	In Vitro Mechanical Properties	Bone Regeneration In Vivo (microCT Analysis)
Karamat-Ullah et al. (2021) [23]	Silica (0.6 or 3 ratio)–silk fibroin gel-based ink for hybrid aerogel-based scaffold conjugated with CM (cecropin melittin)–RGD peptide (60 or 120 μg)	Sol–gel-based hybrid ink, 3D printing, unidirectional freeze casting, freeze drying (−60 °C for 24 h)	Mesopore diameter (nm): 15.6 (Silica-3-SF-CM-RGD-60)–17.2 (Silica–3-SF) (BJH method). Micropore size (μm): 18−20 (nanoCT and SEM imaging). Macropore size (μm): 500–1000 (microCT analysis and SEM imaging).	Young’s modulus (kPa): 31.98 (Silica-0.6-SF-CM-RGD-60)–283.5 (Silica-3-SF-CM-RGD-60) along the pore (freezing) direction	_
Ng et al. (2022) [11]	Methacrylated silk fibroin (SF-MA) and ciprofloxacin-loaded methacrylated hollow mesoporous silica microcapsules (HMSC-MA) aerogel-based composite scaffolds (SF-MA-HMSC)	Self-assembled SF methacrylation, synthesis of HMSC, HMSC methacrylation, sol–gel of HMSC–MA and SF–MA-15 and 30 (4 and 2 *w/v*%), 3D printing of the hydrogel ink, ciprofloxacin-loading, UV photopolymerization/crosslinking, freeze casting (liquid N_2_), freeze drying (−60 °C, 24 h)	Pore size (µm): ~1000 (macropores); ~100–120 (interconnected micropores) (micro and nanoCT analysis); mesoporosities	_	_
Al-Jawuschi et al. (2023) [21]	Silk fibroin methacrylate (SF-MA) incorporated with methacrylate polyvinyl pyrrolidine (PVP)-bismuth sulfide (Bi_2_S_3_) nanobelts 3D aerogel-based composite scaffold loaded with sorafenib (SFN) (SF-MA-20-PVP-Bi_2_S_3_-MA-x) (x = 5, 10 and 15 mass of loaded nanobelts)	Self-assembled SF methacrylation, PVP-Bi_2_S_3_ nanobelts prepared by hydrothermal method, PVP-Bi_2_S_3_ nanobelts methacrylation, sol–gel of SF-MA-20-PVP-Bi_2_S_3_-MA-x, 3D printing of the hydrogel ink, UV photopolymerization/crosslinking, freeze casting (liquid N_2_), freeze drying (−40 °C, 24 h), SFN loading	Pore size (µm): ~1000 (macropores) (SEM imaging); 7–23 (interconnected micropores) (SEM imaging); no meso- or nanopores	_	_
Chen et al. (2021) [9]	Dual network silk fibroin (SF)/cellulose/nHA (S–C–H) composite aerogel	Sol–gel, crosslinking, freeze drying (24 h)	_	Tensile strength (MPa): 7.73 (S–C–H (1:8:1 ratio)). Bending strength (MPa): 25.91 (S–C–H (1:8:1 ratio)).	_
Chen et al. (2022) [25]	Mineralized (hydroxyapatite) silk fibroin (SF)/cellulose (M–S–C) interpenetrating network composite aerogel	Sol–gel, mineralization in situ (24 h), freeze drying (−56 °C, 48 h)	Interconnected (SEM imaging) porosity increased from 98.4% (S–C) to 99.2% (M–S–C) after in situ mineralization (ethanol liquid immersion method)	Compressive strength (MPa): 22.4 (M–S–C), 11.1 (M–C). Elastic modulus (MPa): ~600 (M–S–C)–~375 (M–C and S–C).	_
Liu et al. (2022a) [10]	Poly(lactic acid)/gelatin (PLA/Gel)/silica (SiO_2_ 0, 20, 40, 60%) nanofiber composite aerogel	Electrospinning of PLA/Gel nanofibers and SiO_2_ nanofibers, sol–gel, freezing (−80 °C 12 h), freeze drying (72 h), muffle furnace for crosslinking	PLA/Gel exhibited compact nanofiber sheets along with mesopores; PLA/Gel/SiO_2_ aerogels showed loose fibers morphology and uniform pores with increasing SiO_2_ (SEM imaging)	Ultimate compressive strength (kPa): 516.7 (PLA/Gel/SiO_2_-60)–866.6 (PLA/Gel/SiO_2_-40) (dry state). Compressive modulus (kPa): ~60 (PLA/Gel/SiO_2_-40)–~30 (PLA/Gel/SiO_2_-60) (dry state). Shape recovery rate (wet state) of PLA/Gel/SiO_2_-40 was 94% and 91% after 50 and 100 cycles.	Calvaria defect (diameter 5 mm) in rats—12 weeks. New bone coverage (%): 93 (PLA/Gel/SiO_2_-40), ~60 (PLA/Gel), ~35 (control). BV/TV (%): ~65 (PLA/Gel/SiO_2_-40), ~35 (PLA/Gel), ~30 (control). BMD (g·cm^–3^): 0.213 (PLA/Gel/SiO_2_-40), 0.131 (PLA/Gel), 0.097 (control).
Liu et al. (2022b) [22]	Polyvinyl alcohol (PVA)/modified carbon nanotubes (MCNTs 0.05, 0.10 or 0.15 wt.%)/hydroxyapatite (HAp) aerogel scaffolds	PVA/MCNTs suspension, freezing (liquid N_2_ for 10 min), freeze drying (48 h); suspension of PVA/MCNTs (0.05 wt.%) aerogels in SBF for 3 days for mineralization	Main pore size distribution 1000–1700 nm. Porosity (%): 70.10 (PVA/MCNTs (0.05 wt.%)/HAp)–76.03 (PVA) (mercury porosimetry).	Stiffness (at 70% deformation, MPa): ~1.5 (PVA/MCNTs (0.10 and 0.15 wt.%))–4.2 (PVA/MCNTs (0.05 wt.%)/HAp)	Calvaria defect (diameter 5 mm) in rats—8 weeks. BV/TV and BS/TS (%): ~100% (PVA/MCNTs (0.05 wt.%)/HAp), ~80% (PVA/MCNTs (0.05 wt.%)). BMD (g·cm^–3^): ~0.5 (PVA/MCNTs (0.05 wt.%)/HAp), ~0.4 (PVA/MCNTs (0.05 wt.%)).
Weng et al. (2018) [44]	3D hybrid nanofiber aerogels of PLGA–collagen–gelatin (PCG) and Sr–Cu codoped bioactive glass (BG) nanofibers (60:40) loaded with E7–BMP (bone morphogenetic protein)–2 peptide	Sol–gel, electrospinning, crosslinking; fragmentation of nanofibers, sol–gel of nanofibers, rapid freeze casting (–30 °C, −50 °C or −80 °C in ethanol for 1 min or –196 °C in liquid N_2_), freeze drying (−55 °C for 72 h), thermal crosslinking, solvent exchange, freeze drying	Size of pores was around 30 µm (for freezing temperatures −30 °C to −80 °C). Much smaller pores for −196 °C freezing temperatures (SEM imaging).	Compression modulus (MPa): ~0.25 (PCG–BG (25:75))–~2.25 (PCG–BG (100:0))	Calvaria defect (diameter 8 mm) in rats—8 weeks. BV/TV (%): 65 (PCG–BG (60:40) E7–BMP). Bone formation area (%): 68 (PCG–BG (60:40) E7–BMP).
Li et al. (2021) [43]	3D hybrid nanofiber aerogels of PLGA–collagen–gelatin (PCG) and bioactive glass (BG) nanofibers (60:40) loaded with polycation miR-26a nanoparticles (NPs)	Sol–gel, electrospinning of PCG and BG nanofibers, crosslinking; fragmentation of nanofibers, sol–gel of nanofibers, freezing (−20 °C for 3 h and −80 °C for 15 min), freeze drying, thermal crosslinking, freeze drying	Interconnected pores, diameter 100 μm (SEM imaging)	_	Calvaria defect (diameter 8 mm) in rats—4 weeks. Bone volume (mm3): 2.1 (Blank), 7.5 (aerogel/miR–NC NPs), 21.8 (aerogel/miR-26a NPs). BV/TV (%): 6.0 (Blank), 21.4 (aerogel/miR–NC NPs), 62.2 (aerogel/miR-26a NPs). Bone formation area (%): 7.3 (Blank), 19.7 (aerogel/miR–NC NPs), 56.4 (aerogel/miR-26a NPs).
Ruphuy et al. (2018) [56]	Nano-hydroxyapatite/chitosan (nHApCS, 70/30) hybrid scaffold (different neutralization methods: untreated, NaOHEtOH, scCO_2_-75/75)	nHAp and CS dispersion, freezing (–20 °C overnight), freeze drying (24 h) and: –immersion in NaOH/ethanol, washing, freeze drying or –scCO_2_ (2 cycles at 75 °C, 8.0 MPa) residual solvent removal and sterilization	Total porosity (%): 81 (n–HApCS–scCO_2_-75/75),83 (nHApCS–untreated), 93 (nHApCS–NaOHEtOH) (gas pycnometer). Mean pore size (µm): 86 (nHApCS–untreated), 72 (nHApCS–scCO_2_–75/75), 63 (nHApCS–NaOHEtOH) (SEM imaging).	Storage modulus (at 1 Hz after 1 h in PBS, kPa): 6.8 (nHApCS–untreated), 20.5 (nHApCS–scCO_2_–75/75), 13.3 (nHApCS–NaOHEtOH).	_
Souto-Lopes et al. (2023) [57]	3D nanohydroxyapatite/chitosan (nHAp/CS, 70/30) or CS scaffold	nHAp and CS dispersion, freezing (–20 °C overnight), freeze drying (24 h), scCO_2_ (continuous batch cycles at 75 °C, 8.0 MPa for 2 h) residual solvent removal and sterilization	Total porosity (%): 77 (CS), 78 (nHAp/CS). Full interconnectivity. Pore diameter (µm): 152 (CS), 201 (nHAp/CS) (microCT analysis).	Storage modulus (at 1 Hz, kPa): 37.0 (nHAp/CS 1 h in PBS)–38.8 (nHAp/CS 28 days in PBS); 11.9 (CS 1 h in PBS)–7.8 (CS 28 days in PBS). Storage modulus (at 10 Hz, kPa): 47.1 (nHAp/CS 1 h in PBS)–42.3 (nHAp/CS 28 days in PBS); 16.3 (CS 1 h in PBS)–8.7 (CS 28 days in PBS).	_
Liu et al. (2019) [58]	Graphene oxide (0, 0.05, 0.1, 0.2% GO)–collagen (COL) aerogels	Sol–gel, freezing, freeze drying (−50 °C for 8 h), crosslinking, freeze drying	Porosity (%): 78.1 (0.2% GO–COL)–83.6 (0.1% GO–COL) (liquid displacement method). Pore size (μm): 100–160 (SEM imaging).	Elastic modulus (compression, MPa): 0.20 (COL)–0.51 (0.2% GO–COL)	2 craniofacial bone defects (diameter 5 mm) in rats—12 weeks. BV (mm3): ~3 (0.05% GO–COL)–~6 (0.2% GO–COL). BV/TV (%): ~8 (0.05% GO–COL)–~16 (0.2% GO–COL).
Li et al. (2018) [55]	Sugarcane aerogel-derived borate bioglass scaffolds (SBBS)	Sol–gel preparation of borate glass, curing; freezing (−80 °C for 48 h) of sugarcane carbon hydrogels, freeze drying; borate loading on sugarcane aerogels, oven drying	_	Compressive strength (MPa): ~0.55 (less concentrated curing solution)–~0.75 (more concentrated curing solution) for 30-5B SBBS	Bilateral ulnar bone defect (7 mm radial length and 3 × 3 mm^2^ cross-sectional area) in rabbits—8 weeks. The defect with vertically oriented SBBS was completely healed.
Ye et al. (2019) [16]	Nano-hydroxyapatite/PLLA/gelatin (nHA/PLA/Gel)–peptide (PEP, BMP–2 derived peptides) 3D nanofibrous scaffolds	PLA and Gel solution homogenization, nHA dispersion, electrospinning, cut and dispersion of nanofibers, freeze drying (24 h), thermo-crosslinking, crosslinking, freeze drying (48 h), polydopamine (pDA) coating, immersion in BMP–2 peptide solution, freeze drying	Interconnected pores from tens of microns to 300 µm (SEM imaging)	Young’s modulus (kPa): ~45 (PLA/Gel)–~65 (nHA/PLA/Gel) (wet)	Calvaria defect (diameter 6 mm) in rats—8 weeks. BV/TV (%): ~15 (PLA/Gel)–~45 (nHA/PLA/Gel–PEP).
Zhang et al. (2021) [46]	Three-layered scaffold of poly(L–lactide)/gelatin/hyaluronic acid/chondroitin sulfate (PLA/Gel/HA/CS) fibers and PLA/Gel gradient biomineralized fiber composite aerogels grafted with E7-peptide (A–E7G)	Electrospinning of PLA/Gel/HA/CS and PLA/Gel fibers, porogen incorporation, freezing (liquid N_2_), freeze drying (24 h), crosslinking heat treatment, porogen removal, soaking of PLA/Gel in 5SBF (24 or 48 h), aerogel layer adhesion with photocurable methacrylated gelatin (GelMA), photocrosslinking, E7-peptide grafting, freeze drying		Compressive stress (at 80% strain, MPa): 0.23 (PLA/Gel/HA/CS aerogel layer), 0.62 (PLA/Gel 5SBF 24 h aerogel layer), ~0.6 (A–G trilayered scaffold), 1.4 (PLA/Gel 5SBF 48 h aerogel layer)	Bilateral double knee osteochondral full thickness defects (4 mm · 4 mm) in rabbits—12 weeks. BV/TV (%): ~20 (Blank), ~30 (PLA/Gel), ~35 (A–G) ~50 (A–E7G). Tb.Th (mm): ~0.2 (Blank)–~0.35 (A–E7G). Tb.N (1/mm): ~1.25 (Blank)–~2.0 (A–E7G).

BJH—Barrett–Joyner–Halenda method; SEM—scanning electron microscopy; PBS—phosphate buffer saline; BV/TV—bone volume/tissue volume; BMD—bone mineral density; BS/TS—bone surface/tissue surface; BV—bone volume; SBF—simulated body fluid; Tb.Th—trabecular thickness; Tb.N—trabecular number.

### 2.3. Ambient Pressure Drying

Ambient pressure (or evaporation) drying is a simple, economic technique with lower associated risks [59] for obtaining porous materials (xerogels) from wet-gel solutions [33,59]. While it has been regularly utilized in industrial scale-up [33], however, this technique is slow [59], usually requires the substitution of water by an organic solvent such as ethanol or acetone [33] and typically takes several days to complete [35]. Compared to the drying techniques described in Section 2.1 and Section 2.2, ambient pressure drying frequently leads to a higher shrinkage ratio of the material [33,59] and pore structure collapses due to capillary tensions [28], depending on the precursor solutions and solvents employed [59]. The use of ionic gelation and crosslinking agents also influences the final morphology of the scaffold [59]. Consequently, these materials tend to exhibit higher bulk densities and lower porosities and surface areas, but also more stable mechanical properties [59]. Nevertheless, compared to other aerogels, these ambient dried materials have been successfully employed in various applications. For example, they have shown promise in sustained drug delivery (where the slower delivery rate, compared to highly porous aerogels, is more beneficial in some biomedical applications) and in wound dressing and healing (due to their great absorbability of exudates) [59]. A few bone tissue-engineering applications of composite xerogel scaffolds have also been studied in more recent years, as detailed in Table 3.

**Table 3 materials-16-04483-t003:** Aerogel (xerogel) composite scaffolds prepared by ambient pressure drying.

References	Material	Preparation Methodology	Porous Properties	In Vitro Mechanical Properties	Bone Regeneration In Vivo (microCT Analysis)
Perez-Moreno et al. (2021) [60]	Silica (SiO_2_)–chitosan (CS 8 wt.%)–tricalcium phosphate (TCP 10 or 20 wt.%) (SCS8T) xerogels	Ultrasonic preparation of precursors, ultrasonic sol–gel, washing (unwashed (U), in ethanol for 1 (E1) or 7 days (E7) or in water for 30 d (W30)), ambient pressure drying (80 °C, 48 h)	Pore size (nm): 3.0 (SCS8T20_U)–3.3 (SCS8_U, SCS8T10_U); 4.7 (SCS8T20_E1)–6.7 (SCS8_E1); 6.0 (SCS8T20_E7)–7.1 (SCS8_E7); 2.5 (SCS8_W30)–2.7 (SCS8T10_W30, SCS8T20_W30) (BJH method)	_	_
Perez-Moreno et al. (2023) [53]	Chitosan (CS 8 wt.%)—silica (SiO_2_) hybrid/tricalcium phosphate (TCP 10 wt.%) xerogel (SCS8T10X)	Ultrasonic preparation of precursors, ultrasonic sol–gel, aging and solvent exchange, ambient pressure drying (50 °C)	Pore size (nm): 4.7 (SiO_2_X), 5.0 (SCS8X), 7.5 (SCS8T10X) (BJH method)	_	_

BJH—Barrett–Joyner–Halenda method.

## 3. Properties of Composite Aerogels for Bone Regeneration

One of the most essential features of bone graft biomaterials is the existence of a highly interconnected porous structure, comprising a wide range of pore sizes and geometries, in order to meet the needs of a rough surface for adhesion of osteoblasts and osteoclasts [15,41,61], as well as the circulation of cells, oxygen, nutrients and metabolites and the formation of new vessels [10,25,37]. Moreover, this type of architecture should be balanced with the scaffolds’ ability to withstand compressive stress from the functional demands of bone [10,15,24,61]. Otherwise, if the scaffold does not recover upon compression, its porous configuration is lost and it might jeopardize its osteoconductivity effect in vivo [7,10,15]. Despite the importance of the chosen drying technique, the performance of the hybrid/composite scaffold still depends to a great extent on its composition, including factors such as the type and degree of crosslinking, concentration and molecular weight of polymers and inorganic phases [19,42].

### 3.1. Aerogels’ Porous Structure

When comparing Table 1, Table 2 and Table 3, it is clear that most aerogel composite scaffolds for bone regeneration were prepared through freeze drying instead of the classical supercritical drying or even ambient pressure drying techniques. Generally, biomaterials prepared by scCO_2_ drying tend to exhibit higher porosity (Figure 3) but smaller pore diameters when compared to freeze-dried materials [33]. They usually lack macropores [41] (pores above 100 µm). While in freeze-dried materials the pore diameters tend to range from 20 to 160 µm (micro/macroporosity) [37], scCO_2_-dried biomaterials show high mesoporosity (nanometer range pore size) [41]. Nevertheless, freeze-dried materials may also show mesoporosity and therefore feature aerogel-like properties [28].

**Figure 3 materials-16-04483-f003:**
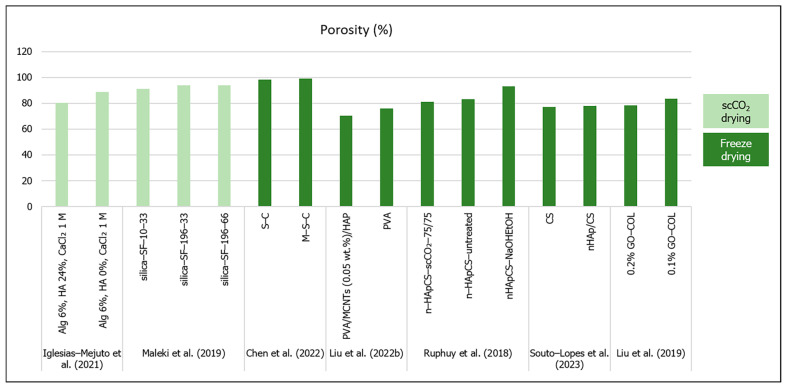
Porosity (%) of aerogel scaffolds prepared through different drying methods (data from [22,25,26,41,56,57,58]).

However, it is important to consider the methodologies used to analyze and quantify the scaffolds’ porosity and pore size, since the limitations of each method can bias these values and the perception of the aerogel structure [18,28]. Therefore, SEM (scanning electron microscopy) imaging, micro and nanoCT analysis are adequate for the assessment of macro and micropores [23,28,56,57,62], while mesoporosity and textural parameters (specific pore volume, pore size distributions and mean pore diameter) are usually determined from the desorption branch of N_2_ sorption isotherm by the Barrett–Joyner–Halenda (BJH) method, and specific surface area is calculated with the Brunauer–Emmett–Teller (BET) five-point method [18,23,32,41,54]. Surface area, volume, density and porosity have also been determined by mercury intrusion [22,62]; helium pycnometer has been used to estimate apparent density and porosity [26,41,56,62]; the ethanol liquid immersion method [25] and liquid displacement method have been used for porosity measurement [58].

In terms of pore size, the studies’ results in Table 1 showed that the pore size scale ranges from 8.6 [54] to 31 nm [41] (mesopores) to a few micrometers [26]. Perez-Moreno et al. (2020) developed silica (SiO_2_)/chitosan (CS 0, 4, 8, 16, 20 wt.%) (SCS) composite aerogels with mesopores ranging from 11 to 14 nm [7]. The authors noticed that there was not a defined pore size behavior with the variations in CS content, but pore interconnectivity was higher for pure SiO_2_ and SCS20 aerogels, which suggested that CS had a distorting influence on the aerogel network [7]. In a 2021 (Table 3) study, Perez-Moreno et al. developed ambient dried silica (SiO_2_)–chitosan (CS 8 wt.%) xerogels incorporating tricalcium phosphate (TCP 10 or 20 wt.%) (SCS8T) to enhance bioactivity [60]. They compared the effect of several washing methods (ethanol or water) before drying and concluded that ethanol washing, in general, was more effective in reducing unreacted chemical residues, and thus exhibited higher mesopore diameters (from 4.7 to 7.1 nm) than in water or no washing (from 2.5 to 3.3 nm) [60]. Moreover, the authors verified that the incorporation of TCP in the ethanol-washed xerogels increased the density by 40 to 50%, while in water-washed xerogels the density increased by only 4 to 7%, showing that TCP was released from the biomaterial by water [60]. Furthermore, the pore diameters tended to decrease with the higher TCP contents in ethanol-washed samples (4.7 nm for SCS8T20_E1 vs. 5.5 for SCS8T10_E1 and 6.0 nm for SCS8T20_E7 vs. 6.3 for SCS8T10_E7) [60]. Furthermore, Perez-Moreno et al.’s (2023) study (Table 1 and Table 3) compared the effect of scCO_2_ and ambient pressure drying on the scaffolds’ properties [53]. Firstly, they came to the conclusion that scCO_2_ drying impaired the TCP incorporation, since it would be leached by the scCO_2_ procedure from the aerogel [53]. They also reported that the xerogel scaffolds showed lower pore diameters (between 4.7 and 7.5 nm) than the aerogels (16.9 to 17.3 nm), and the incorporation of CS 8 wt.% and TCP 10 wt.% increased the pore diameters when compared to both types of pure silica matrices (SiO_2_A and SiO_2_X) [53].

The hybrid silica–gelatin-based aerogels developed by Reyes-Peces et al. (2023) (Table 1) exhibited lower mean pore diameters (in a nanometer scale), specific surface area and total pore volume with increasing contents of gelatin (from 15 to 30% wt.) and crosslinker (organosilaner 3-glycidoxypropyltrimethoxysilane—GPTMS) [54]. The authors attributed these results to a more intertwined hybrid structure, due to higher organic network content. Moreover, these hybrid aerogels did not show a micro or macropore distribution [54]. In a study by Iglesias-Mejuto et al. (2021) 3D-printed alginate–hydroxyapatite (Alg–HA) aerogel scaffolds were obtained after scCO_2_ drying [41]. They compared their scaffolds’ structure properties with and without hydroxyapatite included in the alginate aerogel, and concluded that although the hybrid scaffolds displayed a markedly lower specific surface area and total pore volume, the differences in the mean pore diameter were not meaningful (19 nm for Alg 6%, HA 0%, CaCl2 1 M formulation; 31 nm for Alg 6%, HA 24%, CaCl2 1 M) [41]. The authors compensated the lack of macropores through the scaffold design obtained by 3D printing (with the aligned microfibers separated by microporous gaps), which allowed for reaching high total porosity values (80 to 88%) and a pore size range (from nanopores to macropores) more favorable for bone tissue engineering [41]. When developing scaffolds for bone regeneration, it is desirable to improve the macroporosity to enhance cell migration, neovascularization and mass transfer [32]. Furthermore, the presence of mesoporosity has a relevant impact on implant topography and scaffold bioactivity [35], and high porosity is an essential feature of scaffolds for expanding cells and their interactions in vivo [37,44].

In order to overcome the drawbacks in establishing a hierarchical oriented microstructure composed of macropores as well as mesopores, Maleki et al. (2019) (Table 1) developed a silica–silk fibroin hybrid aerogel prepared through a combination of freeze casting followed by scCO_2_ drying [26]. In that particular case, the hybrid aerogel that exhibited more equilibrated outcomes in general was the one obtained at lower cooling constant temperature and rate (silica–SF-196-33) [26]. This aerogel was the lightest (bulk density 0.075 g/cm^3^) and displayed the highest macro-sized honeycomb-like pores (17.84 µm) and the highest total porosity (94%) [26]. After that study, Karamat-Ullah et al. (2021) (Table 2) developed a silica (0.6 or 3 ratio)–silk fibroin gel-based ink for a 3D-printed hybrid aerogel-based scaffold conjugated with CM (cecropin melittin)–RGD peptide (60 or 120 μg), which showed antimicrobial and cell adhesive properties [23]. However, in that work, the scaffolds’ hierarchically organized porous structure was also evaluated, but instead of using unidirectional freeze casting followed by supercritical drying, they performed freeze casting and drying [23]. Therefore, the aerogel showed mesopores from the sol–gel process, micropores after the freeze casting, and macropores through the 3D printing [23]. In the same research subject, Ng et al. (2022) (Table 2) developed 3D-printed methacrylated silk fibroin (SF–MA) and ciprofloxacin-loaded methacrylated hollow mesoporous silica microcapsules (HMSC–MA) aerogel-based composite scaffolds (SF–MA–HMSC), with hierarchical interconnected porosity [11]. Similarly, pores of around 1000 µm were observed from CAD 3D model and printing, micropores of 100 to 120 µm were formed from freeze casting and drying, and mesoporosity derived from the self-assembled SF methacrylation and photo-crosslinking processes [11]. Finally, based on the 3D-printing methodologies developed by Ng et al. (2022) [11], the authors showed the development of silk fibroin methacrylate (SF-MA) incorporated with methacrylate polyvinyl pyrrolidine (PVP)—bismuth sulfide (Bi_2_S_3_) nanobelts 3D aerogel-based composite scaffold loaded with sorafenib (SFN) (SF-MA-20-PVP-Bi_2_S_3_-MA-x) for photothermally anticancer drug (SFN) release and cancer cell ablation. They reported similar macropores (1000 µm) resulting from the 3D printing but smaller micropores (7 to 23 µm) from unidirectional freeze casting and drying, and absence of mesopores [21].

Weng et al. (2018) produced 3D hybrid nanofiber aerogels of PLGA–collagen–gelatin (PCG) and Sr–Cu codoped bioactive glass (BG) nanofibers at different freezing temperatures, thermal crosslinking conditions and PCG:BG ratios [44]. Their results showed that despite the overall porosity being unaffected, the pore size (from SEM image analysis) was highly influenced by the high freezing temperature gradients, i.e., freezing temperatures between −30 °C and −80 °C did not make differences in pore size (around 30 µm), but when frozen in liquid N_2_, the pore size was so reduced that the aerogel should not be appropriate for bone regeneration [44]. In another study, Li et al. (2021) prepared the PCG–BG (60:40) (without Sr–Cu doping) aerogel by a slower freezing protocol (−20 °C for 3 h and then −80 °C for 15 min), using shorter BG nanofiber segments and different thermal crosslinking conditions (48 °C for 10 min) [43]. As a consequence of the freezing conditions and shorter nanofibers, the scaffolds featured larger interconnected pores with a 100 µm diameter, when compared to the Weng et al. (2018) study [44], which should improve both cell infiltration and vascularization [43].

In an earlier study, Chen et al. (2021) developed a dual-network silk fibroin/cellulose/nHA (S–C–H) aerogel scaffold through freeze drying [9]. Afterwards, they kept the same interpenetrating dual organic phase but tested different in situ mineralization times before freeze drying, instead of dispersing nHA in the sol–gel [25]. Their extremely porous M–S–C aerogel scaffold (mineralized for 24 h) showed significantly higher interconnected porosity (99.2%) than the S–C (non–mineralized) aerogel (98.4%) [25]. Another study by Liu et al. (2022a) produced a composite aerogel by electrospinning PLA/Gel nanofibers and SiO_2_ nanofibers (0 to 60% *w*/*w*), followed by −80 °C freezing and freeze drying [10]. The PLA/Gel/SiO_2_ scaffolds displayed more favorable porous morphologies than the plain PLA/Gel scaffold, which suffered shrinkage upon thermal crosslinking [10]. These studies showed how the reinforcement of the microstructure through the inclusion of an inorganic phase might improve the general porous configuration of the scaffolds.

Additionally, Ruphuy et al. (2018) developed freeze-dried nano-hydroxyapatite/chitosan (n–HapCS, 70/30) hybrid scaffolds with pore size ranging between 50 and 100 µm, using different methods for solvent (acetic acid) neutralization [56]. Their SEM imaging analysis showed that the freeze-dried scaffolds without any further treatment exhibited a favorable interconnected porous structure (86 µm mean pore size, 83% porosity), and the supercritical process kept a similar porous structure (72 µm, 81%) [56]. By contrast, NaOH/ethanol neutralization altered the scaffold’s morphology and decreased the mean pore size (63 µm) but increased the overall porosity (93%), probably due to the second freeze-drying step [56]. Souto-Lopes et al. (2023) tested the freeze-dried nHAp/CS scaffolds processed by continuous supercritical CO_2_, instead of two independent cycles, and the microCT results showed total porosity of 78%, without statistical difference from the CS scaffold (77%) [57]. Moreover, the nHAp/CS scaffolds showed a wider range of pore sizes, though there was no significant difference between the mean pore size of the nHAp/CS (201 µm) and CS (152 µm) scaffolds [57].

Liu et al. (2022b) prepared PVA and PVA/MCNTs (0.05, 0.10 or 0.15 wt.%) freeze-dried aerogels that simulated the natural cuttlebone porous configuration and mimicked its bone structure [22]. Since a fast deep-freeze method (10 min in liquid N_2_) of the suspensions was used, the final aerogels featured a lower pore size (1000 to 1700 nm) [22] when compared to the other previously mentioned freeze-dried scaffolds [11,16,21,23,43,58]. However, these pore sizes did not fall into the mesoporous range. The porosity of the PVA aerogels (76%) diminished with the addition of the inorganic phase (MCNTs) and it was lower (around 70%) for the PVA/MCNTs (0.05 wt.%) and PVA/MCNTs (0.05 wt.%)/HAp [22], which contrasted with the results from Chen et al. (2022) [25]. Ye et al. (2019) incorporated BMP-2 peptides into nano-hydroxyapatite/PLLA/gelatin (nHA/PLA/Gel–PEP) 3D nanofibrous scaffolds prepared by freeze drying of short nanofibers [16]. The produced scaffolds showed a range of pore sizes from tens of microns to 300 µm. The pore sizes were higher in the PLA/Gel scaffolds when compared to nHA/PLA/Gel and nHA/PLA/Gel–PEP, but there were more pores and more uniform distribution in nHA/PLA/Gel and nHA/PLA/Gel–PEP when compared to PLA/Gel scaffolds [16]. The freeze-dried graphene oxide (GO)–collagen (COL) aerogels developed by Liu et al. (2019) did not show significant differences in terms of pore size ranges (100 to 160 µm) and total porosity (around 80%) with different GO ratios (0, 0.05, 0.1, 0.2%), but these values were appropriate for bone regeneration [58].

### 3.2. Aerogels’ In Vitro Mechanical Properties

In both Table 1 and Table 2, the results from the reported mechanical tests were performed under dry conditions, unless stated otherwise. When evaluating the mechanical properties of aerogel scaffolds for bone regeneration, it is important to consider the conditions in which the tests were performed, since there are usually differences in the mechanical performance of these biomaterials in dry or wet conditions, as discussed below.

This material behavior is observed in the study of the hybrid silica–GPTMS–gelatin-based aerogels developed by Reyes-Peces et al. (2023) (Table 1) [54]. The supercritical dried aerogels showed an elastic behavior to compression and higher stiffness in dry conditions with increasing gelatin content (maximum compressive strain for SG25 was 27.67%) and crosslinking agents [54]. However, since the aim of these biomaterials was to be implanted in a living organism with moisture from blood and organic fluids, when tested in wet conditions, the performance of these hybrid aerogels decreased for all compression parameters (though still in the MPa scale), with the lowest stiffness shown for the aerogels with higher organic phase contents [54]. This phenomenon may be explained by the fact that gelatin allowed the swelling of phosphate buffer saline (PBS), which showed the respective swelling ratios of 2.32, 3.42 and 3.04 for 15, 25 and 30% of gelatin content, respectively [54]. Ye et al. (2019) (Table 2) also tested the nHA/PLA/Gel–PEP 3D nanofibrous scaffolds under wet conditions and concluded that the presence of nHA significantly increased the material’s Young’s modulus (~65 kPa) [16]. However, the pDA-assisted coating for the incorporation of BPM-2 peptide significantly decreased the Young’s modulus (~50 kPa). Nonetheless, the range of the Young’s modulus is quite different from that obtained by Reyes-Peces et al. (2023) (1.65 to 3.71 MPa) [54], which showed very low-sized pores (mesoporosity) through scCO_2_ treatment, when compared to the micrometric range pore diameter up to 300 µm obtained after freeze drying [16].

Furthermore, in Ruphuy et al.’s study (2018), the freeze-dried nHAp/CS scaffolds were subjected to dynamic mechanical analysis after 1 h in PBS [56]. Despite all scaffold types exhibiting a fast swelling (24 g g^–1^ in 10 min for the nHApCS–scCO_2_–75/75 scaffold), both the untreated and NaOH/EtOH neutralized-scaffolds showed structural disruption, which impaired their mechanical performance (storage modulus at 1 Hz was ~7 and 13 kPa, respectively) compared to the materials produced with scCO_2_ technology (20.5 kPa) [56]. Nevertheless, all tested scaffolds showed an increase in the storage modulus (from 6.8 to ~15 kPa for nHApCS–untreated, from 13.3 to ~15 kPa for nHApCS–NaOHEtOH and from 20.5 to ~28 kPa for nHApCS–scCO_2_–75/75, at 1 and 10 Hz, respectively) and Tan Delta (loss factor, from ~0.3 to ~0.5 for nHApCS–untreated, from ~0.1 to ~0.3 for nHApCS–NaOHEtOH and from ~0.1 to ~0.4 for nHApCS–scCO_2_–75/75, at 1 and 10 Hz, respectively) [56]. These parameters were also evaluated by Souto-Lopes et al. (2023) [57]. Moreover, the work showed differences in the compressive storage modulus between the scaffolds with (37.0 kPa after 1 h in PBS and 38.8 kPa after 28 days in PBS at 1 Hz; 47.1 kPa after 1 h in PBS and 42.5 kPa after 28 days in PBS at 10 Hz) and without (11.9 kPa after 1 h in PBS and 7.8 kPa after 28 days in PBS at 1 Hz; 16.3 kPa after 1 h in PBS and 8.7 kPa after 28 days in PBS at 10 Hz) nHAp in their composition [57]. The storage modulus results for the nHAp/CS scaffold were approximately three-fold higher (at both 1 and 10 Hz) when compared to the CS material after swelling in PBS and five-fold higher after 28 days of incubation in PBS. Therefore, the increased biodegradation with time after CS scaffold incubation in saline solution explains the biomechanical response of the different biomaterials [57].

When comparing the mechanical strength only in dry conditions, the hybrid silica–gelatin-based aerogels developed by Reyes-Peces et al. (2023) exhibited the best performance [54] (Table 1). The scCO_2_-dried silica/chitosan composite aerogel developed by Perez-Moreno et al. (2020) showed a viscoelastic behavior and a marked decrease in the Young’s modulus with the incorporation of CS (from 11.57 MPa in SiO_2_ to 2.61 MPa in SCS4) [7]. On the other hand, those authors reported that the aerogel with lower CS content (SCS4) exhibited increased compressive strength but decreased toughness, when compared to the pure SiO_2_ aerogel, which was driven by the loss of pore interconnectivity, as referred to in a previous section (Section 3.1) [7]. Water absorption occurred in a linear way for aerogels with and without CS, being expected a swelling behavior of the CS network [7]. The scCO_2_-dried hybrid silica–silk fibroin aerogel developed by Maleki et al. (2019) with more favorable structural parameters in general (silica–SF-196-33) also featured the highest maximum strength (1.6 MPa) and Young’s modulus (7.3 MPa) [26]. In order to improve the hierarchical porous structure of the aerogels, Karamat-Ullah et al. (2021) showed that freeze casting and drying instead of scCO_2_ drying (Maleki et al. (2019) [26]) changed the scaffolds’ porous structure and influenced the scaffolds’ stiffness (compression test), since their Young’s modulus decreased significantly from a range of 4.03–7.3 MPa [26] to 31.98–283.5 kPa [23].

When evaluating the mechanical performance considering the organic/inorganic ratio of the freeze-dried aerogels, Chen et al. (2022) observed that the M–S–C composite aerogel showed the highest compressive strength results (22.4 MPa) after 24 h of in situ mineralization (a comparative study along times of 1, 3, 6 and 12 h of in situ mineralization) [25]. Moreover, despite there being no differences in the microstructural parameters between the freeze-dried GO–COL aerogels developed by Liu et al. (2019), the compression elastic modulus increased with GO contents (value for 0.2% GO–COL was significantly higher than that for 0, 0.05 and 0.1% of GO) [58]. These authors also determined that 0.1% and 0.2% GO–COL scaffolds absorbed more than 1500% of water (about 1.27-fold and 1.35-fold more than the COL scaffold, respectively), showing that increasing GO content contributed to the increase in materials’ hydrophilicity [58]. However, higher inorganic phase content could not necessarily mean higher mechanical strength, as shown by Liu et al. (2022a) with their study of PLA/Gel/SiO_2_ (0 to 60% *w*/*w*) scaffolds [10]. In that case, the PLA/Gel/SiO_2_-40 aerogel displayed significantly higher mechanical properties when compared to the other aerogels tested with different SiO_2_ contents (PLA/Gel/SiO_2_-60 showed lower compressive strength when compared to the PLA/Gel–control) [10]. Moreover, the PVA/MCNT (0.05 wt.%) freeze-dried aerogels developed by Liu et al. (2022b) also showed higher stiffness (3.5 MPa at 70% strain) than the PVA/MCNTs (0.1 wt.%) and PVA/MCNTs (0.15 wt.%) (~1.5 MPa for both), due to higher content of MCNTs that jeopardizes their own particle dispersion in the aerogel [22]. Nevertheless, the stiffness of PVA/MCNTs (0.05 wt.%) increased with the addition of HAp after 3 days of SBF mineralization (~4.0 to 4.2 MPa) [22]. By contrast, Weng et al. (2018) observed an exponential increase in the Young’s modulus with the increase in the polymer content of the PCG–BG aerogel [44]. They showed the highest results with PCG 100%, after optimal thermal crosslinking at 52 °C for 10 min [44]. Even though their initial objective was to incorporate the highest possible ratio of Sr–Cu codoped BG nanofibers, in order to achieve the best outcomes in terms of osteoinduction and neovascularization, the opposite in terms of the materials’ mechanical properties was observed [44].

In terms of biomimetic strategies for aerogel scaffold development, Li et al. (2018) prepared sugarcane-derived borate bioglass (tetraethyl orthosilicate (TEOS), tributyl borate (TBB), triethyl phosphate (TEP)) aerogel scaffolds by freeze drying with anisotropic properties due to the multilevel structure of the sugarcane, in order to match the internal structure of natural bone [55]. The aerogel prepared with a ratio of two of TEOS to TBB (named 30-5B) showed more favorable biodegradation and bioactivity properties and, after reinforcement with different concentrations of a phosphate curing solution, the composite showed the highest compression strength (~0.75 MPa) with higher concentration of curing solution when compared with the composite prepared with the lowest concentration of curing solution (~0.55 MPa) [55]. Another study by Zhang et al. (2021) developed an aerogel-based three-layered scaffold (A–G) made of electrospun fibers (poly(L–lactide)/gelatin/hyaluronic acid/chondroitin sulfate—PLA/Gel/HA/CS) using the aerogel technology for osteochondral regeneration [46]. The scaffolds were subsequently freeze dried, crosslinked, gradient biomineralized and grafted with E7 peptide (A–E7G) [46]. The upper (chondral) layer (A–C) composed of PLA/Gel/HA/CS aerogel showed the lowest compressive resistance (0.23 MPa), followed by the middle (intermediate calcified cartilage zone) layer (A–M) made of PLA/Gel aerogel soaked in 5SBF for 24 h (0.62 MPa), as well as the three-layered scaffold (A–G, ~0.6 MPa) and crosslinked PLA/Gel aerogel (A–U, ~0.6 MPa) [46]. The highest compressive stress was observed with PLA/Gel aerogel that was biomineralized for 48 h (A–B, 1.4 MPa) referring to the bottom (osseous) layer [46].

### 3.3. Aerogels and In Vivo Bone Regeneration Potential

The final goal of the development of aerogel scaffolds for bone regeneration is their capacity for guiding bone cell migration and inducing differentiation of host mesenchymal stem cells into osteoblasts, in order to secrete collagen fibers and promote new bone formation in a tissue defect [23]. At this stage, every property of a biomaterial, from the composition and morphology to the mechanical and biocompatibility performance, is an important aspect that may contribute or not to promote bone regeneration [26]. Figure 4 summarizes the most relevant results of in vivo bone regeneration from most of the studies analyzed below in this section.

From the studies detailed in Table 1 (aerogel scaffolds prepared by scCO_2_ drying), Maleki et al. (2019) reported that their silica–SF-196-33 scaffold (which exhibited more appropriate microstructural architecture and mechanical properties) was chosen to be tested in vivo in a rat femur defect model [26]. Despite the low macropore mean diameter (17.84 µm), but high porosity (94%), after 25 days, the microCT analysis revealed that bone density in the defect containing the implanted scaffold was similar to that of native bone [26]. Moreover, no evidence of inflammation, negative tissue response or systemic toxicity was observed [26].

Several freeze-dried aerogels (Table 2) have already been tested in vivo in critical--size bone defects. Liu et al. (2022b) observed that the PVA/MCNTs (0.05 wt.%)/HAp scaffold’s BV/TV and BS/TS was nearly 100% (the defect was almost fully filled by new bone tissue) after 8 weeks in rat calvaria model [22]. All bone formation quantifications measured by microCT were significantly higher for the scaffold with HAp when compared to the other groups (empty control defect, PVA and PVA/MCNTs—0.05 wt.%), due to the osteoinduction triggered by the presence of the HAp nanoparticles [22]. Liu et al. (2022a) observed that the PLA/Gel/SiO_2_-40 scaffold showed increasing BV/TV, BMD and new bone formation (93% after 12 weeks in rat calvaria model) with time, with significant differences compared to the plain PLA/Gel and the empty controls at all time points [10]. On the other hand, Liu et al. (2019) showed lower bone formation after 12 weeks post implantation in rat calvaria model [58]. According to these authors’ conclusions, the 0.1 and 0.2% GO–COL groups showed a >1.5-fold higher BV (~6 mm^3^) and BV/TV ratio (~16%) when compared to the other two groups [58].

There were also some published results that described the inclusion of osteoinductive molecules in the proposed aerogel’s composition, in order to improve their regeneration outcomes. Weng et al. (2018) performed a bone implant of PCG–BG (60:40) scaffolds and compared the mineralized tissue formation in rat calvaria defects with and without the introduction E7–BMP peptide [44]. After 8 weeks, there was no difference in terms of bone volume and bone formation area when compared the PCG–BG (60:40) scaffolds and the empty defects (~25% and ~30%, respectively), but the PCG–BG (60:40) E7–BMP scaffold showed 65% bone formation inside the defect and 68% of new bone area [44]. Li et al. (2021) incorporated nanoparticles (NPs) of microRNAs-26a (miR-26a) in the PCG–BG (60:40) aerogels to promote bone mesenchymal stem cell osteogenic differentiation and tissue vascularization [43]. In terms of in vivo bone regeneration in rat calvaria defects at the 4th week of implantation, aerogel with miR–NC NPs (negative control—without miR-26a) promoted significantly more bone formation (7.5 mm^3^ of bone volume, 21.4% of BV/TV, 19.7% of bone formation area) than that observed in the empty defect (2.1 mm^3^ of bone volume, 6.0% of BV/TV, 7.3% of bone formation area), and the addition of the miR-26a significantly increased the bone regeneration (21.8 mm^3^ of bone volume, 62.2% of BV/TV, 56.4% of bone formation area) [43]. However, despite the larger pore diameters in the Li et al. (2021) study (100 µm) [43], the bone regeneration results were similar to the two types of PCG–BG (60:40) aerogels prepared in the two different studies [43,44].

Another study involving BMP-2-derived peptides was conducted by Ye et al. (2019), which stated that the presence of nHA significantly increased the in vivo new bone formation in rat calvaria defects implanted with nHA/PLA/Gel 3D nanofibrous scaffolds when compared to the control material, but the presence of BMP-2 peptides further increased the BV/TV results (~45%) [16]. Moreover, after 12 weeks, Zhang et al. (2021) showed that although both three-layered aerogel scaffolds (A–G and A–E7G) for osteochondral regeneration tended to increase new bone formation, only the scaffolds modified with E7-peptide showed significant differences (BV/TV ~50%; Tb.Th ~0.35 mm; Tb.N ~2.0 mm^–1^) when compared to the control defects [46]. Furthermore, upon nanoindentation tests, the regenerated osteochondral tissue in the A–E7G aerogel-implanted defect exhibited significantly higher reduced modulus (~20 MPa) and hardness (~400 kPa) when compared with the other tested groups [46].

An interesting approach was reported by Li et al. (2018) with the 30-5B sugarcane-derived borate bioglass aerogel scaffold with anisotropic properties [55]. They implanted their scaffold in either a horizontal or vertical orientation of the sugarcane microstructure in bilateral ulnar bone defects in rabbits [55]. After 8 weeks, the vertically oriented aerogel scaffolds that were parallel to the long axis of the rabbit ulna showed complete bone formation along the defect, when compared to the control and horizontally oriented scaffold, which showed lower tissue formation [55].

## 4. Future Research Approaches for Aerogels and Bone Regeneration

Despite the recent advances in aerogel technology for biomedical applications, particularly for bone regeneration, there are still some issues that must be overcome in order to successfully obtain reliable clinical results with these graft materials. Robust production methods must be further developed to obtain aerogels comprising both mesopores and micro/macropores, which might be accomplished by combining several techniques such as 3D printing, freeze casting and freeze drying [11,23]. It would also be interesting to further develop anisotropic scaffolds with oriented porous structures matching the surrounding native bone [55,63].

Mainly, there is also the need to improve the balance between high porosity and mechanical strength of composite aerogels, specifically under wet conditions. The plasticizing effect of aqueous solutions in the polymeric matrices is very significant, and it is necessary to increase the stiffness after swelling. Further testing of crosslinking conditions/agents, as well as nanofiber technologies (namely the use of short nanofibers) could improve the materials’ properties, mechanically reinforcing the aerogel structure [16,43,44].

Additionally, reducing the use of solvents and chemicals used in the scaffolds’ synthesis would improve the biocompatibility and decrease the production costs and the environmental impact. Moreover, there is a need to perform more studies regarding the incorporation of important biological molecules to provide higher osteoinduction properties and to boost host cellular response and tissue regeneration [16,43,44].

Another research line in the bone bioengineering technologies is the development of tissue regeneration based on in vitro 3D tissue models (organ-on-a-chip) instead of using in vivo animal experiments. By using microfluidic systems, this approach overcomes the limitations of simplistic 2D cell cultures [1], while avoiding the obstacles related to ethical issues and high costs associated with animal experiments [64]. Although significant efforts have already been made in the development of organ-on-a-chip technology for the gastrointestinal tract, vasculature, lung, kidney and others, there are not yet so many effective examples of bone-on-a-chip models [64,65]. In fact, most of the bone-on-a-chip devices produced so far were developed to study diseases such as osteoporosis or cancer/metastasis and related therapies [66], but those for bone tissue development or regeneration are scarce [64]. Microfluidic systems allow the simulation of biomechanical cues that are important for bone cell functionality through shear stresses caused by the fluid flow [1,65]. Both natural and synthetic-based organic/inorganic compounds have been employed for the obtention of 3D hydrogels for bone-on-a-chip devices [1,65]. Three-dimensional bioprinting has also proved to be useful in developing these devices [1,64,66,67], as well as other scaffold preparation methods frequently employed for bone regeneration [66]. Additionally, co-culture cells, such as primary human osteoblasts (HOBs) [6] or human bone marrow-derived mesenchymal stem cells (hBM–MSCs) and human umbilical vein endothelial cells (HUVECs) [68], human adipose-derived stem cells (ADSCs) [69] or metastatic breast cancer (BrCa) cells [70], and joining specific growth factors such as BMP-2 could be employed to better mimic bone microenvironment [68] in either physiological or pathological conditions.

## 5. Conclusions

The present review compared the preparation methods of composite aerogels for bone regeneration and the outcomes in terms of their porous structure, mechanical performance, swelling and in vivo bone repair in small animal models. Having in mind the recent changes in the aerogel definition, it was noticeable that most authors have been preparing their aerogel-based bone scaffolds through freeze drying instead of supercritical and ambient pressure drying, due to the need for achieving a wider range of pore sizes, including micro/macropores. Supercritical drying technology may produce scaffolds with very high total porosities, but mainly exhibit mesoporosity, which, despite providing a favorable texture and high surface area for cell attachment, may not be so effective for vascularization, cell proliferation and migration and molecular transport. Other production methods such as printing and freeze casting, should be used more to overcome these drawbacks. On the other hand, supercritical dried aerogels seemed to provide better mechanical compressive strength in dry conditions when compared with similar freeze-dried composite aerogels, though other important key factors such as the precursors (organic and inorganic), crosslinking agent and technologies such as electrospinning seemed to play relevant roles in the mechanical performance. However, the mechanical tests in wet conditions deserve to be further studied, because they should provide more reliable viscoelastic properties after swelling. This review highlighted the performance of aerogels towards the relevance of the inorganic phase, as well as the presence of osteoinductive molecules such as BPM-2-derived peptides for new bone formation in vivo in critical-size defects. Though microCT technology enabled reducing the number of animals used for assessing in vivo bone regeneration quantification, research in this field should focus on improving knowledge in areas such as 3D in vitro bone regeneration models for replacing/reducing live animal experimentation, in order to avoid unethical procedures.

## Figures and Tables

**Figure 1 materials-16-04483-f001:**
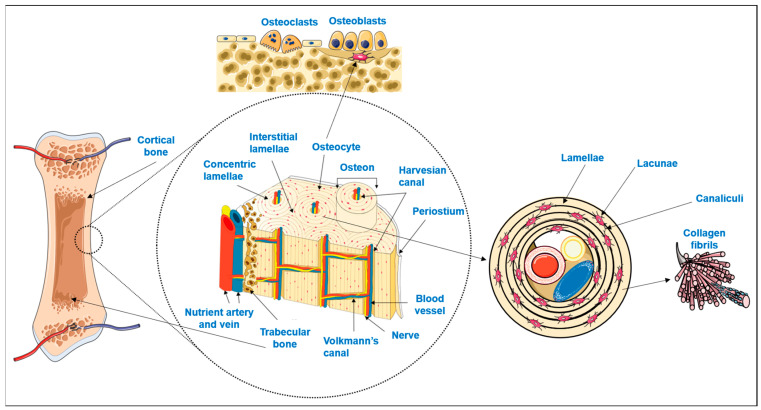
Schematic representation of bone hierarchical macro, micro and nanostructure and main cellular populations (adapted from [5]). Parts of the figure were drawn by using pictures from Servier Medical Art. Servier Medical Art by Servier is licensed under a Creative Commons Attribution 3.0 Unported License (https://creativecommons.org/licenses/by/3.0/).

**Figure 2 materials-16-04483-f002:**
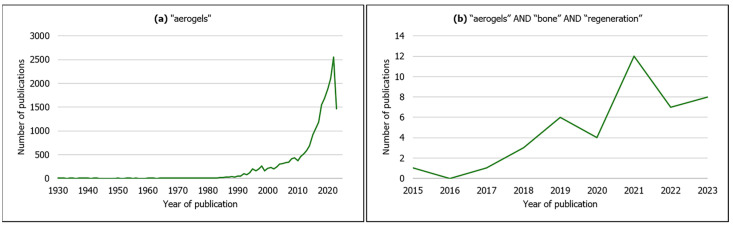
Evolution in the number of publications by year of publication on (**a**) aerogels from all scientific fields (search keyword: “aerogels”); (**b**) aerogels for bone regeneration (search keywords: “aerogels” AND “bone” AND “regeneration”). Source: Scopus (search date: 9 June 2023).

**Figure 4 materials-16-04483-f004:**
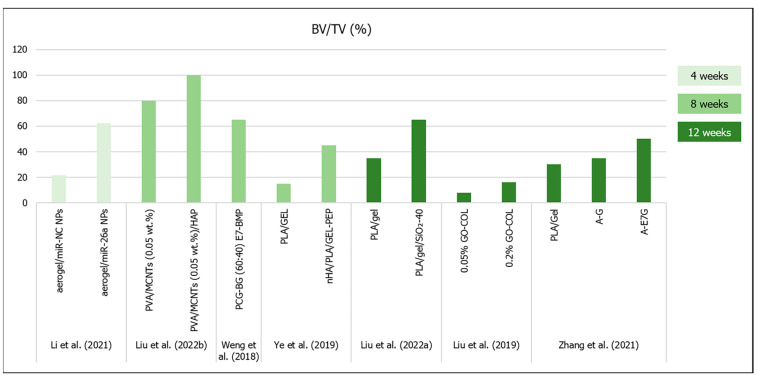
Results (bone volume/tissue volume (BV/TV %)) from bone regeneration in vivo (obtained by microCT analysis) with the implantation of aerogel scaffolds of different materials (data from [10,16,22,43,44,46,58]). All scaffolds were implanted in rat calvaria model, except for the study from ref. [45] (rabbit knee).

## Data Availability

The data are unavailable.

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
