# Peer review of "Bioengineering Composite Aerogel-Based Scaffolds That Influence Porous Microstructure, Mechanical Properties and In Vivo Regeneration for Bone Tissue Application"

_materials, 2023, doi:10.3390/ma16124483_

Round 1

Reviewer 1 Report

Minor editing of English language required.

Author Response

Please, see attached file.

Reviewer 2 Report

The present review by Mariana Souto-Lopes et al. reports on the latest advances in aerogel-based scaffolds for bone tissue application on last 5 years. The review is well organized and contains up-to-date information. I recommend to accept this work after minor revision. My comments are listed below.

Page 3 line 101: m2 .g-1 should be corrected

Page 3 line 103: for my opinion this sentence should list the most commonly used polymers for aerogel- based scaffolds preparation. It also applies to line 186: the mentioning polymers should be given in the text.

The definition and comparison of macro/micro/meso pores should be given.

As the presented manuscript is a review article some additional figures should be include.For example, the comparison of the porous structure of some discussed materials.

Page 11 line 242: is the term “osteoconductibility” correct? The authors should clarify it.

Some minor changes should be done:

“-“ replaced by “–“

Celsius degree icon needs to be corrected

Author Response

Please, see attached file.

Reviewer 3 Report

- The authors review papers reporting the use of supercritical and freeze-drying aerogels. There are no references to works that involve materials dried at atmospheric pressure. Historically, such materials are called xerogels, but currently the difference between xerogels and aerogels is blurring, so I would expect a mention of this group of materials as well what the authors themselves indicate in lines 161-165. Thus, the xerogel materials for bone tissue aplications should also been mentioned

The review is available e.g. here: https://www.mdpi.com/2310-2861/8/6/334

- Tables 1 and 2 would be clearer if the references were in a separate column and not linked to the material. If there is not enough space, you can consider horizontal layout of the pages containing these tables.

Author Response

Please, see attached file.
